# RESIDUAL GATED GRAPH CONVNETS

## ABSTRACT

Graph-structured data such as social networks, functional brain networks, gene regulatory networks, communications networks have brought the interest in generalizing deep learning techniques to graph domains. In this paper, we are interested to design neural networks for graphs with variable length in order to solve learning problems such as vertex classification, graph classification, graph regression, and graph generative tasks. Most existing works have focused on recurrent neural networks (RNNs) to learn meaningful representations of graphs, and more recently new convolutional neural networks (ConvNets) have been introduced. In this work, we want to compare rigorously these two fundamental families of architectures to solve graph learning tasks. We review existing graph RNN and ConvNet architectures, and propose natural extension of LSTM and ConvNet to graphs with arbitrary size. Then, we design a set of analytically controlled experiments on two basic graph problems, i.e. subgraph matching and graph clustering, to test the different architectures. Numerical results show that the proposed graph ConvNets are 3-17% more accurate and 1.5-4x faster than graph RNNs. Graph ConvNets are also 36% more accurate than variational (non-learning) techniques. Finally, the most effective graph ConvNet architecture uses gated edges and residuality. Residuality plays an essential role to learn multi-layer architectures as they provide a 10% gain of performance.

## 1 INTRODUCTION

Convolutional neural networks of LeCun et al. (1998) and recurrent neural networks of Hochreiter & Schmidhuber (1997) are deep learning architectures that have been applied with great success to computer vision (CV) and natural language processing (NLP) tasks. Such models require the data domain to be regular, such as 2D or 3D Euclidean grids for CV and 1D line for NLP. Beyond CV and NLP, data does not usually lie on regular domains but on heterogeneous graph domains. Users on social networks, functional time series on brain structures, gene DNA on regulatory networks, IP packets on telecommunication networks are a a few examples to motivate the development of new neural network techniques that can be applied to graphs. One possible classification of these techniques is to consider neural network architectures with fixed length graphs and variable length graphs.

In the case of graphs with fixed length, a family of convolutional neural networks has been developed on spectral graph theory by Chung (1997). The early work of Bruna et al. (2013) proposed to formulate graph convolutional operations in the spectral domain with the graph Laplacian, as an analogy of the Euclidean Fourier transform as proposed by Hammond et al. (2011). This work was extended by Henaff et al. (2015) to smooth spectral filters for spatial localization. Defferrard et al. (2016) used Chebyshev polynomials to achieve linear complexity for sparse graphs, Levie et al. (2017) applied Cayley polynomials to focus on narrow-band frequencies, and Monti et al. (2017b) dealt with multiple (fixed) graphs. Finally, Kipf & Welling (2017) simplified the spectral convnets architecture using 1-hop filters to solve the semi-supervised clustering task. For related works, see also the works of Bronstein et al. (2017b), Bronstein et al. (2017a) and references therein.

For graphs with variable length, a generic formulation was proposed by Gori et al. (2005); Scarselli et al. (2009) based on recurrent neural networks. The authors defined a multilayer perceptron of a vanilla RNN. This work was extended by Li et al. (2016) using a GRU architecture and a hidden state that captures the average information in local neighborhoods of the graph. The work of Sukhbaatar et al. (2016) introduced a vanilla graph ConvNet and used this new architecture to solve learning

communication tasks. Marcheggiani & Titov (2017) introduced an edge gating mechanism in graph ConvNets for semantic role labeling. Finally, Bruna & Li (2017) designed a network to learn non-linear approximations of the power of graph Laplacian operators, and applied it to the unsupervised graph clustering problem. Other works for drugs design, computer graphics and vision are presented by Duvenaud et al. (2015); Boscaini et al. (2016); Monti et al. (2017a).

In this work, we study the two fundamental classes of neural networks, RNNs and ConvNets, in the context of graphs with arbitrary length. Section 2 reviews the existing techniques. Section 3 presents the new graph NN models. Section 4 reports the numerical experiments.

## 2 NEURAL NETWORKS FOR GRAPHS WITH ARBITRARY LENGTH

### 2.1 RECURRENT NEURAL NETWORKS

**Generic formulation.** Consider a standard RNN for word prediction in natural language processing. Let $h_i$ be the feature vector associated with word $i$ in the sequence. In a regular vanilla RNN, $h_i$ is computed with the feature vector $h_j$ from the previous step and the current word $x_i$, so we have:

$$h_i = f_{\text{VRNN}} ( x_i , \{h_j : j = i - 1\} )$$

The notion of neighborhood for regular RNNs is the previous step in the sequence. For graphs, the notion of neighborhood is given by the graph structure. If $h_i$ stands for the feature vector of vertex $i$, then the most generic version of a feature vector for a graph RNN is

$$h_i = f_{\text{G-RNN}} ( x_i , \{h_j : j \to i\} ) \tag{1}$$

where $x_i$ refers to a data vector and $\{h_j : j \to i\}$ denotes the set of feature vectors of the neighboring vertices. Observe that the set $\{h_j\}$ is unordered, meaning that $h_i$ is intrinsic, i.e. invariant by vertex re-indexing (no vertex matching between graphs is required). Other properties of $f_{\text{G-RNN}}$ are locality as only neighbors of vertex $i$ are considered, weight sharing, and such vector is independent of the graph length. In summary, to define a feature vector in a graph RNN, one needs a mapping $f$ that takes as input an unordered set of vectors $\{h_j\}$, i.e. the feature vectors of all neighboring vertices, and a data vector $x_i$, Figure 1(a).

We refer to the mapping $f_{\text{G-RNN}}$ as the neighborhood transfer function in graph RNNs. In a regular RNN, each neighbor as a distinct position relatively to the current word (1 position left from the center). In a graph, if the edges are not weighted or annotated, neighbors are not distinguishable. The only vertex which is special is the center vertex around which the neighborhood is built. This explains the generic formulation of Eq. (1). This type of formalism for deep learning for graphs with variable length is described by Scarselli et al. (2009); Gilmer et al. (2017); Bronstein et al. (2017a) with slightly different terminology and notations.

**Graph Neural Networks of Scarselli et al. (2009).** The earliest work of graph RNNs for arbitrary graphs was introduced by Gori et al. (2005); Scarselli et al. (2009). The authors proposed to use a vanilla RNN with a multilayer perceptron to define the feature vector $h_i$:

$$h_i = f_{\text{G-VRNN}} (x_i, \{h_j : j \to i\}) = \sum_{j \to i} \mathcal{C}_{\text{G-VRNN}}(x_i, h_j) \tag{2}$$

with

$$\mathcal{C}_{\text{G-VRNN}}(x_i, h_j) = A\sigma(B\sigma(Ux_i + Vh_j)),$$

and $\sigma$ is the sigmoid function, $A, B, U, V$ are the weight parameters to learn.

Minimization of Eq. (2) does not hold a closed-form solution as the dependence computational graph of the model is not a directed acyclic graph (DAG). Scarselli et al. (2009) proposed a fixed-point iterative scheme: for $t = 0, 1, 2, ...$

$$h_i^{t+1} = \sum_{j \to i} \mathcal{C}(x_i, h_j^t), \quad h_i^{t=0} = 0 \ \forall i. \tag{3}$$

The iterative scheme is guaranteed to converge as long as the mapping is contractive, which can be a strong assumption. Besides, a large number of iterations can be computational expensive.

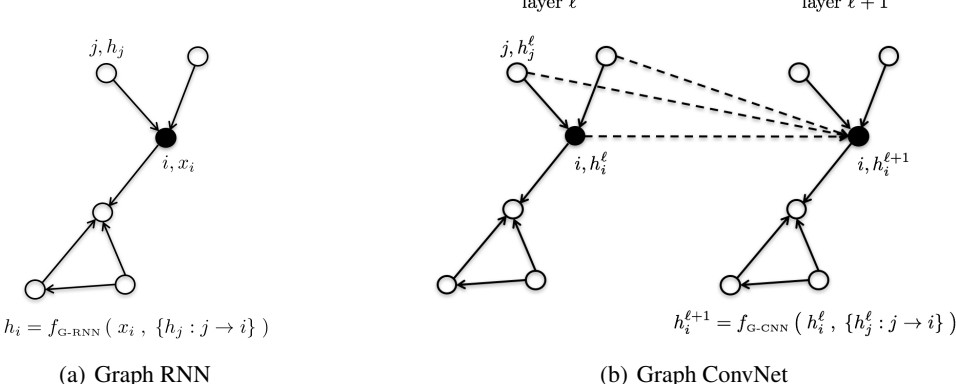

$$h_i = f_{\text{G-RNN}}(x_i, \{h_j : j \to i\})$$

$$h_i^{\ell+1} = f_{\text{G-CNN}}(h_i^\ell, \{h_j^\ell : j \to i\})$$

(a) Graph RNN

(b) Graph ConvNet

Figure 1: Generic feature representation $h_i$ of vertex $i$ on a graph RNN (a) and a graph convNet (b).

**Gated Graph Neural Networks of Li et al. (2016).** In this work, the authors use the gated recurrent units (GRU) of Chung et al. (2014):

$$h_i = f_{\text{G-GRU}}(x_i, \{h_j : j \to i\}) = \mathcal{C}_{\text{G-GRU}}(x_i, \sum_{j \to i} h_j) \tag{4}$$

As the minimization of Eq. (4) does not have an analytical solution, Li et al. (2016) designed the following iterative scheme:

$$h_i^{t+1} = \mathcal{C}_{\text{G-GRU}}(h_i^t, \bar{h}_i^t), \quad h_i^{t=0} = x_i \ \forall i,$$
$$\text{where} \quad \bar{h}_i^t = \sum_{j \to i} h_j^t,$$

and $\mathcal{C}_{\text{G-GRU}}(h_i^t, \bar{h}_i^t)$ is equal to

$$z_i^{t+1} = \sigma(U_z h_i^t + V_z \bar{h}_i^t)$$
$$r_i^{t+1} = \sigma(U_r h_i^t + V_r \bar{h}_i^t)$$
$$\tilde{h}_i^{t+1} = \tanh\big(U_h(h_i^t \odot r_i^{t+1}) + V_h \bar{h}_i^t\big)$$
$$h_i^{t+1} = (1 - z_i^{t+1}) \odot h_i^t + z_i^{t+1} \odot \tilde{h}_i^{t+1},$$

where $\odot$ is the Hadamard point-wise multiplication operator. This model was used for NLP tasks by Li et al. (2016) and also in quantum chemistry by Gilmer et al. (2017) for fast organic molecule properties estimation, for which standard techniques (DFT) require expensive computational time.

**Tree-Structured LSTM of Tai et al. (2015).** The authors extended the original LSTM model of Hochreiter & Schmidhuber (1997) to a tree-graph structure:

$$h_i = f_{\text{T-LSTM}}(x_i, \{h_j : j \in C(i)\}) = \mathcal{C}_{\text{T-LSTM}}(x_i, h_i, \sum_{j \in C(i)} h_j), \tag{5}$$

where $C(i)$ refers the set of children of node $i$. $\mathcal{C}_{\text{T-LSTM}}(x_i, h_i, \sum_{j \in C(i)} h_j)$ is equal to

$$\bar{h}_i = \sum_{j \in C(i)} h_j$$
$$i_i = \sigma(U_i x_i + V_i \bar{h}_i)$$
$$o_i = \sigma(U_o x_i + V_o \bar{h}_i)$$
$$\tilde{c}_i = \tanh\big(U_c x_i + V_c \bar{h}_i\big)$$
$$f_{ij} = \sigma(U_f x_i + V_f h_j)$$
$$c_i = i_i \odot \tilde{c}_i + \sum_{i \in C(i)} f_{ij} \odot c_j$$
$$h_i = o_i \odot \tanh(c_i)$$

Unlike the works of Scarselli et al. (2009); Li et al. (2016), Tree-LSTM does not require an iterative process to update its feature vector $h_i$ as the tree structure is also a DAG as original LSTM. Consequently, the feature representation (5) can be updated with a recurrent formula. Nevertheless, a tree is a special case of graphs, and such recurrence formula cannot be directly applied to arbitrary graph structure. A key property of this model is the function $f_{ij}$ which acts as a gate on the edge from neighbor $j$ to vertex $i$. Given the task, the gate will close to let the information flow from neighbor $j$ to vertex $i$, or it will open to stop it. It seems to be an essential property for learning systems on graphs as some neighbors can be irrelevant. For example, for the community detection task, the graph neural network should learn which neighbors to communicate (same community) and which neighbors to ignore (different community). In different contexts, Dauphin et al. (2017) added a gated mechanism inside the regular ConvNets in order to improve language modeling for translation tasks, and van den Oord et al. (2016) considered a gated unit with the convolutional layers after activation, and used it for image generation.

## 2.2 CONVOLUTIONAL NEURAL NETWORKS

**Generic formulation.** Consider now a classical ConvNet for computer vision. Let $h_{ij}^{\ell}$ denote the feature vector at layer $\ell$ associated with pixel $(i, j)$. In a regular ConvNet, $h_{ij}^{\ell+1}$ is obtained by applying a non linear transformation to the feature vectors $h_{i'j'}^{\ell}$ for all pixels $(i', j')$ in a neighborhood of pixel $(i, j)$. For example, with $3 \times 3$ filters, we would have:

$$h_{ij}^{\ell+1} = f_{\text{CNN}}^{\ell} \left( \{ h_{i'j'}^{\ell} : |i - i'| \leq 1 \text{ and } |j - j'| \leq 1 \} \right)$$

In the above, the notation $\{ h_{i'j'}^{\ell} : |i - i'| \leq 1 \text{ and } |j - j'| \leq 1 \}$ denote the concatenation of all feature vectors $h_{i'j'}^{\ell}$ belonging to the $3 \times 3$ neighborhood of vertex $(i, j)$. In ConvNets, the notion of neighborhood is given by the euclidian distance. As previously noticed, for graphs, the notion of neighborhood is given by the graph structure. Thus, the most generic version of a feature vector $h_i$ at vertex $i$ for a graph ConvNet is

$$h_i^{\ell+1} = f_{\text{G-CNN}} \left( h_i^{\ell} , \{ h_j^{\ell} : j \to i \} \right) \tag{6}$$

where $\{ h_j^{\ell} : j \to i \}$ denotes the set of feature vectors of the neighboring vertices. In other words, to define a graph ConvNet, one needs a mapping $f_{\text{G-CNN}}$ taking as input a vector $h_i^{\ell}$ (the feature vector of the center vertex) as well as an unordered set of vectors $\{ h_j^{\ell} \}$ (the feature vectors of all neighboring vertices), see Figure 1(b). We also refer to the mapping $f_{\text{G-CNN}}$ as the neighborhood transfer function. In a regular ConvNet, each neighbor as a distinct position relatively to the center pixel (for example 1 pixel up and 1 pixel left from the center). As for graph RNNs, the only vertex which is special for graph ConvNets is the center vertex around which the neighborhood is built.

**CommNets of Sukhbaatar et al. (2016).** The authors introduced one of the simplest instantiations of a graph ConvNet with the following neighborhood transfer function:

$$h_i^{\ell+1} = f_{\text{G-VCNN}}^{\ell} \left( h_i^{\ell} , \{ h_j^{\ell} : j \to i \} \right) = \text{ReLU} \left( U^{\ell} h_i^{\ell} + V^{\ell} \sum_{j \to i} h_j^{\ell} \right), \tag{7}$$

where $\ell$ denotes the layer level, and ReLU is the rectified linear unit. We will refer to this architecture as the vanilla graph ConvNet. Sukhbaatar et al. (2016) used this graph neural network to learn the communication between multiple agents to solve multiple tasks like traffic control.

**Syntactic Graph Convolutional Networks of Marcheggiani & Titov (2017).** The authors proposed the following transfer function:

$$h_i^{\ell+1} = f_{\text{S-GCN}}^{\ell} \left( \{ h_j^{\ell} : j \to i \} \right) = \text{ReLU} \left( \sum_{j \to i} \eta_{ij} \odot V^{\ell} h_j^{\ell} \right) \tag{8}$$

where $\eta_{ij}$ act as edge gates, and are computed by:

$$\eta_{ij} = \sigma \left( A^{\ell} h_i^{\ell} + B^{\ell} h_j^{\ell} \right). \tag{9}$$

These gated edges are very similar in spirit to the Tree-LSTM proposed in Tai et al. (2015). We believe this mechanism to be important for graphs, as they will be able to learn what edges are important for the graph learning task to be solved.

## 3 MODELS

**Proposed Graph LSTM.** First, we propose to extend the Tree-LSTM of Tai et al. (2015) to arbitrary graphs and multiple layers:

$$h_i^{\ell+1} = f_{\text{G-LSTM}}^\ell \left( x_i^\ell, \{h_j^\ell : j \to i\} \right) = \mathcal{C}_{\text{G-LSTM}}(x_i^\ell, h_i^\ell, \sum_{j \to i} h_j^\ell, c_i^\ell) \tag{10}$$

As there is no recurrent formula is the general case of graphs, we proceed as Scarselli et al. (2009) and use an iterative process to solve Eq. (10): At layer $\ell$, for $t = 0, 1, ..., T$

$$
\begin{aligned}
\bar{h}_i^{\ell,t} &= \sum_{j \to i} h_j^{\ell,t}, \\
i_i^{\ell,t+1} &= \sigma(U_i^\ell x_i^\ell + V_i^\ell \bar{h}_i^{\ell,t}) \\
o_i^{\ell,t+1} &= \sigma(U_o^\ell x_i^\ell + V_o^\ell \bar{h}_i^{\ell,t}) \\
\tilde{c}_i^{\ell,t+1} &= \tanh\left(U_c^\ell x_i^\ell + V_c^\ell \bar{h}_i^{\ell,t}\right) \\
f_{ij}^{\ell,t+1} &= \sigma(U_f^\ell x_i^\ell + V_f^\ell h_j^{\ell,t}) \\
c_i^{\ell,t+1} &= i_i^{\ell,t+1} \odot \tilde{c}_i^{\ell,t+1} + \sum_{j \to i} f_{ij}^{\ell,t+1} \odot c_j^{\ell,t+1} \\
h_i^{\ell,t+1} &= o_i^{\ell,t+1} \odot \tanh(c_i^{\ell,t+1})
\end{aligned}
$$

$$
\begin{aligned}
\text{and initial conditions:} \quad h_i^{\ell,t=0} &= c_i^{\ell,t=0} = 0, \ \ \forall i, \ell \\
x_i^\ell &= h_i^{\ell-1,T}, \ x_i^{\ell=0} = x_i, \ \ \forall i, \ell
\end{aligned}
$$

In other words, the vector $h_i^{\ell+1}$ is computed by running the model from $t = 0, .., T$ at layer $\ell$. It produces the vector $h_i^{\ell,t=T}$ which becomes $h_i^{\ell+1}$ and also the input $x_i^{\ell+1}$ for the next layer. The proposed Graph LSTM model differs from Liang et al. (2016); Peng et al. (2017) mostly because the cell $\mathcal{C}_{\text{G-LSTM}}$ in these previous models is not iterated over multiple times $T$, which reduces the performance of Graph LSTM (see numerical experiments on Figure 4).

**Proposed Gated Graph ConvNets.** We leverage the vanilla graph ConvNet architecture of Sukhbaatar et al. (2016), Eq.(7), and the edge gating mechanism of Marcheggiani & Titov (2017), Eq.(8), by considering the following model:

$$h_i^{\ell+1} = f_{\text{G-GCNN}}^\ell \left( h_i^\ell, \{h_j^\ell : j \to i\} \right) = \text{ReLU} \left( U^\ell h_i^\ell + \sum_{j \to i} \eta_{ij} \odot V^\ell h_j^\ell \right) \tag{11}$$

where $h_i^{\ell=0} = x_i, \forall i$, and the edge gates $\eta_{ij}$ are defined in Eq. (9). This model is the most generic formulation of a graph ConvNet (because it uses both the feature vector $h_i^\ell$ of the center vertex and the feature vectors $h_j^\ell$ of neighboring vertices) with the edge gating property.

**Residual Gated Graph ConvNets.** In addition, we formulate a multi-layer gated graph ConvNet using residual networks (ResNets) introduced by He et al. (2016). This boils down to add the identity operator between successive convolutional layers:

$$h_i^{\ell+1} = f^\ell \left( h_i^\ell, \{h_j^\ell : j \to i\} \right) + h_i^\ell. \tag{12}$$

As we will see, such multi-layer strategy work very well for graph neural networks.

## 4 EXPERIMENTS

### 4.1 SUBGRAPH MATCHING

We consider the subgraph matching problem presented by Scarselli et al. (2009), see Figure 2(a). The goal is to find the vertices of a given subgraph $P$ in larger graphs $G_k$ with variable sizes.

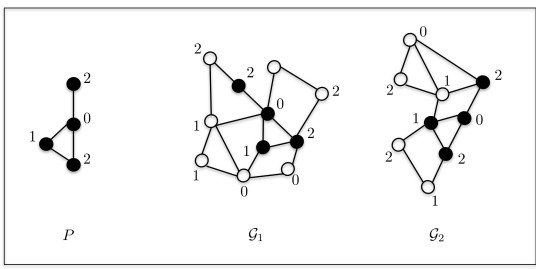 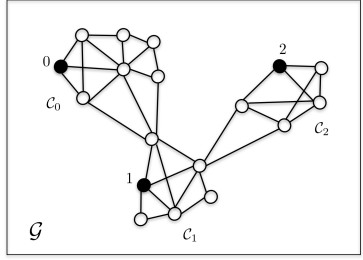

(a) Subgraph matching                    (b) Semi-supervised clustering

Figure 2: Graph learning tasks.

Identifying similar localized patterns in different graphs is one of the most basic tasks for graph neural networks. The subgraph $P$ and larger graph $G_k$ are generated with the stochastic block model (SBM), see for example Abbe (2017). A SBM is a random graph which assigns communities to each node as follows: any two vertices are connected with the probability $p$ if they belong to the same community, or they are connected with the probability $q$ if they belong to different communities. For all experiments, we generate a subgraph $P$ of 20 nodes with a SBM $q = 0.5$, and the signal on $P$ is generated with a uniform random distribution with a vocabulary of size 3, i.e. $\{0, 1, 2\}$. Larger graphs $G_k$ are composed of 10 communities with sizes randomly generated between 15 and 25. The SBM of each community is $p = 0.5$. The value of $q$, which acts as the noise level, is $0.1$, unless otherwise specified. Besides, the signal on $G_k$ is also randomly generated between $\{0, 1, 2\}$. Inputs of all neural networks are the graphs with variable size, and outputs are vertex classification vectors of input graphs. Finally, the output of neural networks are simple fully connected layers from the hidden states.

All reported results are averaged over 5 trails. We run 5 algorithms; Gated Graph Neural Networks of Li et al. (2016), CommNets of Sukhbaatar et al. (2016), SyntacticNets of Marcheggiani & Titov (2017), and the proposed Graph LSTM and Gated ConvNets from Section 3. We upgrade the existing models of Li et al. (2016); Sukhbaatar et al. (2016); Marcheggiani & Titov (2017) with a multilayer version for Li et al. (2016) and using ResNets for all three architectures. We also use the batch normalization technique of Ioffe & Szegedy (2015) to speed up learning convergence for our algorithms, and also for Li et al. (2016); Sukhbaatar et al. (2016); Marcheggiani & Titov (2017). The learning schedule is as follows: the maximum number of iterations, or equivalently the number of randomly generated graphs with the attached subgraph is 5,000 and the learning rate is decreased by a factor $1.25$ if the loss averaged over 100 iterations does not decrease. The loss is the cross-entropy with 2 classes (the subgraph $P$ class and the class of the larger graph $G_k$) respectively weighted by their sizes. The accuracy is the average of the diagonal of the normalized confusion matrix w.r.t. the cluster sizes (the confusion matrix measures the number of nodes correctly and badly classified for each class). We also report the time for a batch of 100 generated graphs. The choice of the architectures will be given for each experiment. All algorithms are optimized as follow. We fix a budget of parameters of $B = 100K$ and a number of layers $L = 6$. The number of hidden neurons $H$ for each layer is automatically computed. Then we manually select the optimizer and learning rate for each architecture that best minimize the loss. For this task, Li et al. (2016); Sukhbaatar et al. (2016); Marcheggiani & Titov (2017) and our gated ConvNets work well with Adam and learning rate $0.00075$. Graph LSTM uses SGD with learning rate $0.075$. Besides, the value of inner iterative steps $T$ for graph LSTM and Li et al. (2016) is 3.

The first experiment focuses on shallow graph neural networks, i.e. with a single layer $L = 1$. We also vary the level of noise, that is the probability $q$ in the SBM that connects two vertices in two different communities (the higher $q$ the more mixed are the communities). The hyper-parameters are selected as follows. Besides $L = 1$, the budget is $B = 100K$ and the number of hidden neurons $H$ is automatically computed for each architecture to satisfy the budget. First row of Figure 3 reports the accuracy and time for the five algorithms and for different levels of noise $q = \{0.1, 0.2, 0.35, 0.5\}$. RNN architectures are plotted in dashed lines and ConvNet architectures in solid lines. For shallow networks, all RNN architectures (graph LSTM and Li et al. (2016)) performs much better, but they also take more time than the graph ConvNets architectures we propose, as well as Sukhbaatar et al.

(2016); Marcheggiani & Titov (2017). As expected, all algorithms performances decrease when the noise increases.

The second experiment demonstrates the importance of having multiple layers compared to shallow networks. We vary the number of layers $L = \{1, 2, 4, 6, 10\}$ and we fix the number of hidden neurons to $H = 50$. Notice that the budget is not the same for all architectures. Second row of Figure 3 reports the accuracy and time w.r.t. $L$ (middle figure is a zoom in the left figure). All models clearly benefit with more layers, but RNN-based architectures see their performances decrease for a large number of layers. The ConvNet architectures benefit from large $L$ values, with the proposed graph ConvNet performing slightly better than Sukhbaatar et al. (2016); Marcheggiani & Titov (2017). Besides, all ConvNet models are faster than RNN models.

In the third experiment, we evaluate the algorithms for different budgets of parameters $B = \{25K, 50K, 75K, 100K, 150K\}$. For this experiment, we fix the number of layers $L = 6$ and the number of neurons $H$ is automatically computed given the budget $B$. The results are reported in the third row of Figure 3. For this task, the proposed graph ConvNet best performs for a large budget, while being faster than RNNs.

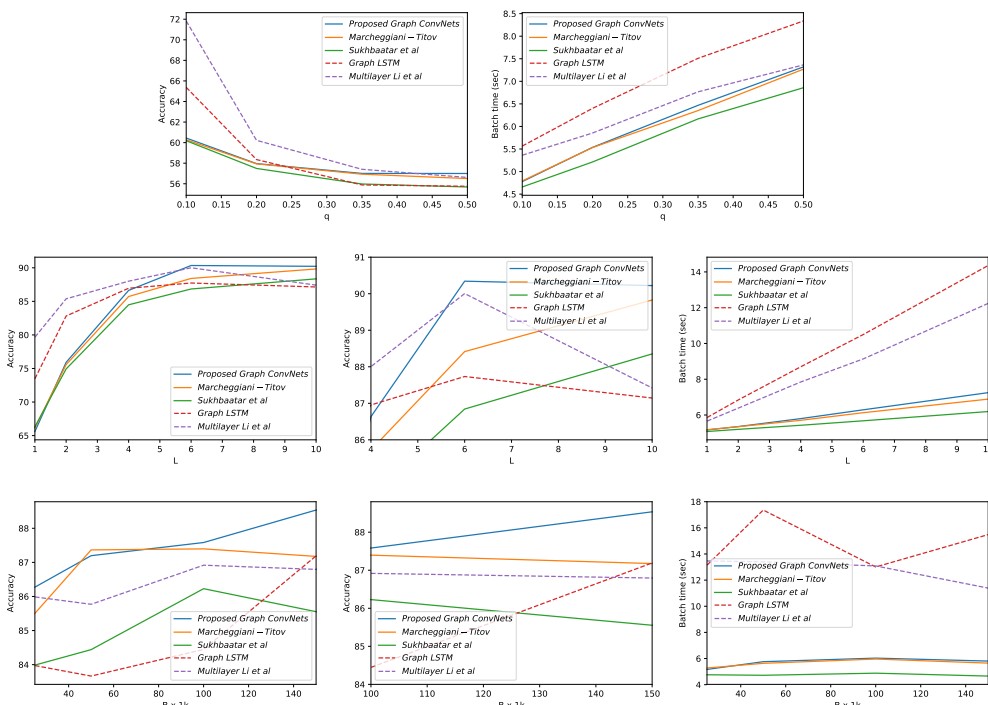

Figure 3: Subgraph matching: First row studies shallow networks w.r.t. noise. Second row investigates multilayer graph networks. Third row reports graph architectures w.r.t. budget.

We also show the influence of hyper-parameter $T$ for Li et al. (2016) and the proposed graph LSTM. We fix $H = 50$, $L = 3$ and $B = 100K$. Figure 4 reports the results for $T = \{1, 2, 3, 4, 6\}$. The $T$ value has an undesirable impact on the performance of graph LSTM. Multi-layer Li et al. (2016) is not really influenced by $T$. Finally, the computational time naturally increases with larger $T$ values.

## 4.2 SEMI-SUPERVISED CLUSTERING

In this section, we consider the semi-supervised clustering problem, see Figure 2(b). This is also a standard task in network science. For this work, it consists in finding 10 communities on a graph given 1 single label for each community. This problem is more discriminative w.r.t. to the architectures than the previous single pattern matching problem where there were only 2 clusters to find (i.e. 50% random chance). For clustering, we have 10 clusters (around 10% random chance). As in the

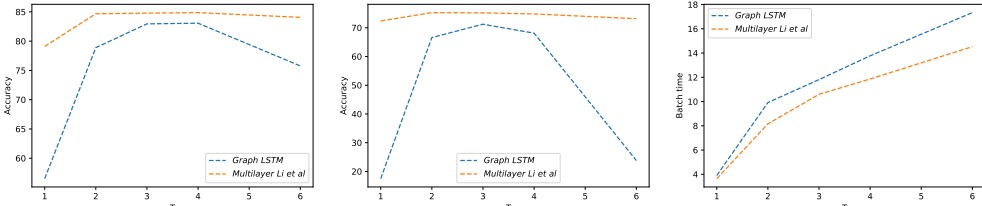

Figure 4: Influence of hyper-parameter $T$ on RNN architectures. Left figure is for graph matching, middle figure for semi-supervised clustering, and right figure are the batch time for the clustering task (same trend for matching).

previous section, we use SBM to generate graphs of communities with variable length. The size for each community is randomly generated between 5 and 25, and the label is randomly selected in each community. Probability $p$ is 0.5, and $q$ depends on the experiment. For this task, Li et al. (2016); Sukhbaatar et al. (2016); Marcheggiani & Titov (2017) and the proposed gated ConvNets work well with Adam and learning rate 0.00075. Graph LSTM uses SGD with learning rate 0.0075. The value of $T$ for graph LSTM and Li et al. (2016) is 3.

The same set of experiments as in the previous task are reported in Figure 5. ConvNet architectures get clearly better than RNNs when the number of layers increase (middle row), with the proposed Gated ConvNet outperforming the other architectures. For a fixed number of layers $L = 6$, our graph ConvNets and Marcheggiani & Titov (2017) best perform for all budgets, while paying a reasonable computational cost.

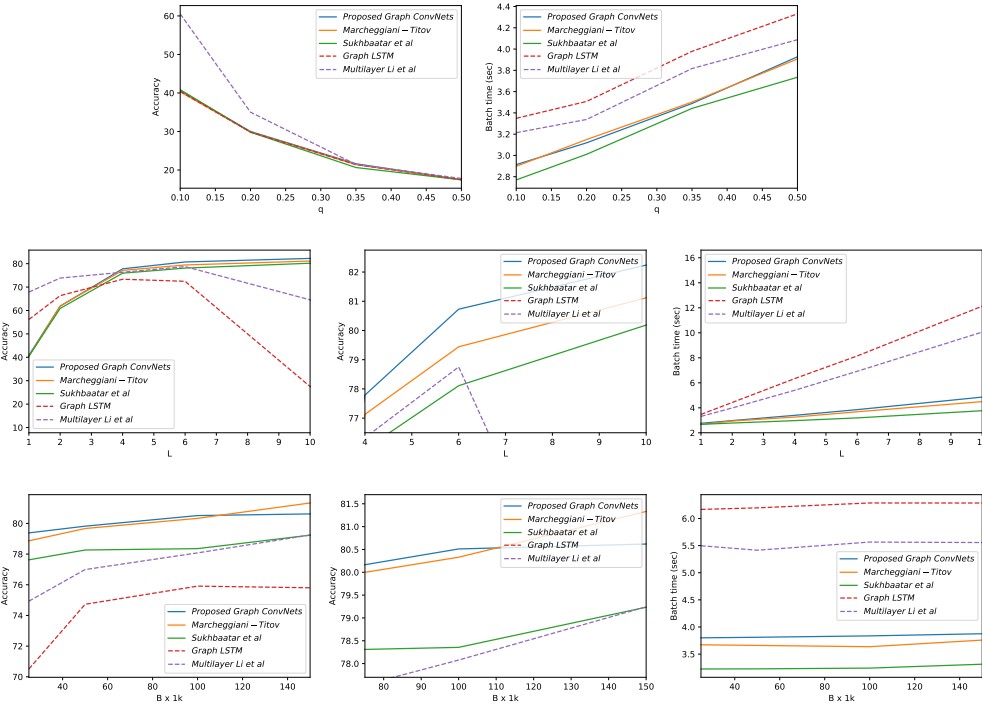

Figure 5: Semi-supervised clustering: First row reports shallow networks w.r.t. noise $q$. Second row shows multilayer graph networks w.r.t. $L$. Third row is about graph architectures w.r.t. budget $B$.

Next, we report the learning speed of the models. We fix $L = 6$, $B = 100K$ with $H$ being automatically computed to satisfy the budget. Figure 6 reports the accuracy w.r.t. time. The ConvNet architectures converge faster than RNNs, in particular for the semi-supervised task.

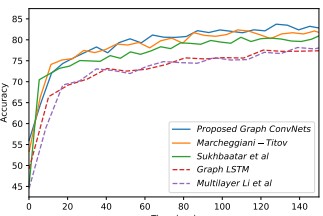 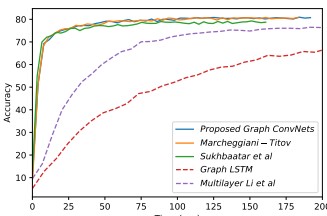

Figure 6: Learning speed of RNN and ConvNet architectures. Left figure is for graph matching and right figure semi-supervised clustering.

To close this study, we are interested in comparing learning based approaches to non-learning variational ones. To this aim, we solve the variational Dirichlet problem with labeled and unlabelled data as proposed by Grady (2006). We run 100 experiments and report an average accuracy of 45.37% using the same setting as the learning techniques (one label per class). The performance of the best learning model is 82%. Learning techniques produce better performances with a different paradigm as they use training data with ground truth, while variational techniques do not use such information. The downside is the need to see 2000 training graphs to get to 82%. However, when the training is done, the test complexity of these learning techniques is $O(E)$, where $E$ is the number of edges in the graph. This is an advantage over the variational Dirichlet model that solves a sparse linear system of equations with complexity $O(E^{3/2})$, see Lipton et al. (1979).

## 5 CONCLUSION

This work explores the choice of graph neural network architectures for solving learning tasks with graphs of variable length. We developed analytically controlled experiments for two fundamental graph learning problems, that are subgraph matching and graph clustering. Numerical experiments showed that graph ConvNets had a monotonous increase of accuracy when the network gets deeper, unlike graph RNNs for which performance decreases for a large number of layers. This led us to consider the most generic formulation of gated graph ConvNets, Eq. (11). We also explored the benefit of residuality for graphs, Eq. (12). Without residuality, existing graph neural networks are not able to stack more than a few layers. This makes this property essential for graph neural networks, which receive a 10% boost of accuracy when more than 6 layers were stacked. Future work will focus on solving domain-specific problems in chemistry, physics, and neuroscience.

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
