# OpenReview forum: "Residual Gated Graph ConvNets"
_ICLR.cc/2018/Conference — Reject_

### Official Review · AnonReviewer1 · 2017-11-27
**Interesting approach that should be better presented**

**Rating:** 7
**Confidence:** 4

**Review:**

The authors revised the paper according to all reviewers suggestions, I am satisfied with the current version.

Summary: this works proposes to employ recurrent gated convnets to solve graph node labeling problems on arbitrary graphs. It build upon several previous works, successively introducing convolutional networks, gated edges convnets on graphs, and LSTMs on trees. The authors extend the tree LSTMs formulation to perform graph labeling on arbitrary graphs, merge convnets with residual connections and edge gating mechanisms. They apply the 2 proposed models to 3 baselines also based on graph neural networks on two problems: sub-graph matching (expressing the problem of sub-graph matching as a node classification problem), and semi supervised clustering.

Main comments:
It would strengthen the paper to also compare all these network learning based approaches to variational ones. For instance, to a spectral clustering method for the semi supervised clustering, or
solving the combinatorial Dirichlet problem as in Grady: random walks for image segmentation, 2006.

The abstract and the conclusion should be revised, they are very vague.
- The abstract should be self contained and should not contain citations.
- The authors should clarify which problem they are dealing with.
- instead of the "numerical result show the performance of the new model", give some numerical results here, otherwise, this sentence is useless.
- we propose ... as propose -> unclear: what do you propose?


Minor comments:
- You should make sentences when using references with the author names format. Example: ... graph theory, Chung (1997) -> graph theory by Chung (1997)
- As Eq 2 -> As the minimization of Eq 2 (same with eq 4)
- Don't start sentences with And, or But

---

> ### Author Response · Authors · 2017-12-23
> **Authors feedback**
>
> We are thankful to the reviewer for her/his comments and time. We hope our answers will clarify the importance of this work and the referee will be inclined to improve her/his evaluation score.
>
> Q: Compare learning based approaches to variational ones
> A: We solved the combinatorial Dirichlet problem with labeled and unlabelled data using [Grady’06, Random walks for image segmentation, Eq. 11, Section B]. The average accuracy (over 100 experiments) for this variational technique is 45.37% (we remind that only 1 label per class is used, and random choice is around 5-15%), while the performance of the best learning technique is 82%. Learning techniques produce better performances with a different paradigm as they use training data with ground truth, while variational techniques do not use such information. The downside is the need to see 2000 training graphs to get to 82%. However, when the training is done, the test complexity of these learning techniques is O(E), where E is the number of edges in the graph. This is an advantage over the variational Dirichlet model that solves a sparse linear system of equations with complexity O(E^1.5) [Lipton-Rose-Tarjan’79]. We thank the referee for this useful comment. We added this comment in the paper.
>
> Q: Abstract, conclusion should be revised
> A: We revised the abstract and conclusion.
>
> Q: The authors should clarify which problem they are dealing with
> A: The general problem we want to solve is learning meaningful representations of graphs with variable length using either ConvNet or RNN architectures. These graph representations can be applied to different tasks such as vertex classification (in this paper for graph matching and graph clustering) and also graph classification, graph regression, graph visualization, graph generative model, etc. We added this comment in the paper.
>
> Q: Give some numerical results
> A: Here is the summary of the results:
> 1. Sub-graph matching:
>   (a) Accuracy of shallow graph NNs is 79% for RNNs and 67% for the proposed ConvNet.
>   (b) Accuracy of deep graph NNs (L=10) is 87% for RNNs and 90% for the proposed ConvNet.
> 2. Semi-supervised graph clustering:
>   (a) Accuracy of shallow graph NNs is 69% for RNNs and 41% for the proposed ConvNet.
>   (b) Accuracy of deep graph NNs (L=10) is 65% for RNNs and 82% for the proposed ConvNet.
> 3. Computational times for graph RNNs is 1.5-4x slower than the proposed ConvNet.
> We added these results in the abstract.
>
> Q: Minor comments
> A: Thank you. We revised the paper accordingly.

---

> > ### Author Response · Authors · 2018-01-08
> > **Authors**
> >
> > We would like to thank the referee for her/his time reviewing the revised paper and for improving her/his evaluation score.

---

### Official Review · AnonReviewer2 · 2017-11-27
**Residual Gated Graph ConvNets**

**Rating:** 3
**Confidence:** 4

**Review:**

The paper proposes an adaptation of existing Graph ConvNets and evaluates this formulation on a several existing benchmarks of the graph neural network community. In particular, a tree structured LSTM is taken and modified. The authors describe this as adapting it to general graphs, stacking, followed by adding edge gates and residuality.

My biggest concern is novelty, as the modifications are minor. In particular, the formulation can be seen in a different way. As I see it, instead of adapting Tree LSTMs to arbitary graphs, it can be seen as taking the original formulation by Scarselli and replacing the RNN by a gated version, i.e. adding the known LSTM gates (input, output, forget gate). This is a minor modification. Adding stacking and residuality are now standard operations in deep learning, and edge-gates have also already been introduced in the literature, as described in the paper.

A second concern is the presentation of the paper, which can be confusing at some points. A major example is the mathematical description of the methods. When reading the description as given, one should actually infer that Graph ConvNets and Graph RNNs are the same thing, which can be seen by the fact that equations (1) and (6) are equivalent.

Another example, after (2), the important point to raise is the difference to classical (sequential) RNNs, namely the fact that the dependence graph of the model is not a DAG anymore, which introduces cyclic dependencies.

Generally, a clear introduction of the problem is also missing. What are the inputs, what are the outputs, what kind of problems should be solved? The update equations for the hidden states are given for all models, but how is the output calculated given the hidden states from variable numbers of nodes of an irregular graph?

The model has been evaluated on standard datasets with a performance, which seems to be on par, or a slight edge, which could probably be due to the newly introduced residuality.

A couple of details :

- the length of a graph is not defined. The size of the set of nodes might be meant.

- at the beginning of section 2.1 I do not understand the reference to word prediction and natural language processing. RNNs are not restricted to NLP and I think there is no need to introduce an application at this point.

- It is unclear what does the following sentence means: "ConvNets are more pruned to deep networks than RNNs"?

- What are "heterogeneous graph domains"?

---

> ### Author Response · Authors · 2017-12-23
> **Authors feedback 1**
>
> We thank the reviewer for her/his time and comments. We provide below specific answers to the questions. We hope the reviewer will update positively her/his decision in view of our answers.
>
> Q: My biggest concern is novelty
> A: Several techniques for graph NNs have been published in the last two years. None of the existing works compare with rigorous numerical experiments which type of architectures (RNNs or ConvNets) should be used for graphs with variable length. The main contribution and novelty of this work is to answer this fundamental question, and give the reader the winning architecture. By running controlled numerical experiments on two basic graph analysis tasks, sub-graph matching and semi-supervised clustering, we reached the conclusion that ConvNets architectures should be used, and the best formulation of graph ConvNets uses edge gates and residuality. We believe such result to be important for future models in this domain (and also a bit controversial) as most graph NNs published in the literature focused on RNN architectures.
>
> Q: Adding stacking and residuality are now standard operations
> A: When we started this work, we doubted that stacking and residuality were helpful for the class of graphs with variable length. Graphs are different data than images: arbitrary graph structures are irregular (e.g. molecule graphs or gene networks), graph convolutional operations are not shift-invariant, and multi-scale structures depend on graph topology. Our original motivation was to numerically study the stacking and residuality properties for graph RNNs and ConvNets, and see if it would be useful.
> We found out the important result that without residuality, *none* of the existing graph NNs can stack more than 2 layers. They simply do not work; they are not able to learn good representations to solve the matching and clustering tasks. Hence, although residuality is quite common in computer vision tasks, our experiments showed that this property is even *more* important for graphs than for images. Quantitatively, we got a boost by at least 10% of accuracy when we stacked more than 6 layers. So, it seemed to us and to other researchers to be a useful result for future research in this domain (that includes applications in chemistry, physics, neuroscience).
>
> Q: The model has been evaluated on standard datasets with a performance, which seems to be on par, or a slight edge, which could probably be due to the newly introduced residuality.
> A: Residuality plays indeed an essential role for graph learning tasks. Without residuality, the existing techniques such as Li-etal, Sukhbaatar-etal, Marcheggiani-Titov are far behind (more than 10% - they actually do not benefit much from multiple layers) than the proposed gated graph residual model. Note that in the experiments, we *did* upgrade the existing techniques with residuality. We could have simply reported the (lower) performances of the original methods, which would have been more impressive on the plots for our model but also not informative.
> The proposed graph ConvNet actually offers a slight improvement compared to Sukhbaatar-etal and Marcheggiani-Titov *when* these models are upgraded with residuality. However, the paper is not an application paper (we do not claim any SOTA on any benchmark dataset), but rather an investigation paper where we want to convey the message that, after rigorous numerical experiments, graph ConvNet architectures should be preferred when we want to design deep learning techniques on arbitrary graphs such as for drugs design.
>
> Q: Graph ConvNets and Graph RNNs are the same thing, equations (1) and (6) are equivalent.
> A: We disagree with this comment - equations 1 and 6 are as different as standard RNNs are distinct from standard ConvNets. The purpose of the mathematical formulations 1 and 6 is to generalize standard ConvNets and RNNs not only to image domains but to arbitrary graph domains (1 and 6 reduce to original RNNs and ConvNets for regular grids). Figure 1 illustrates the fundamental difference between both graph architectures.
>
> Q: An introduction of the problem is missing. What kind of problems should be solved? What are the inputs, the outputs?
> A: The general problem we want to solve is learning meaningful representations of graphs with variable length using either ConvNet or RNN architectures. These graph representations can be applied to different tasks such as vertex classification (for graph matching and clustering in this work) and also graph classification, graph regression, graph visualization, and graph generative model.
> In this work, inputs are graphs with variable size and outputs are vertex classification vectors of input graphs. We added this answer in the paper.

---

> > ### Author Response · Authors · 2017-12-23
> > **Authors feedback 2**
> >
> > Q: Taking original Scarselli and replacing the RNN by LSTM gates
> > A: Yes, this is what we did and explained in Section 3. We do not claim any major contribution for graph LSTM. Our goal was to compare all graph RNN architectures (GRU and LSTM) vs. graph ConvNet architectures. As graph LSTM was not available in the literature, we simply used Scarselli-etal and Tai-etal to extend LSTM to arbitrary graphs.
> >
> > Q: A second concern is the presentation of the paper, which can be confusing at some points.
> > A: We improved the abstract, conclusion and revised some parts of the paper in view of the reviewer’s questions.
> >
> > Q: After (2), the important point to raise is the fact that the dependence graph of the model is not a DAG anymore
> > A: Agreed - we added this comment in the paper.
> >
> > Q: How is the output calculated given the hidden states from variable numbers of nodes of an irregular graph?
> > A: The output is a simple fully connected layer from the convolutional graph features. We added this comment in the paper.
> >
> > Q: The length of a graph is not defined
> > A: Beginning of Sections 4.1 and 4.2 explained how the graph size is designed for each experiment. For graph matching, the size varies randomly between 170 and 270 nodes, and for graph clustering the length is between 50 and 250.
> >
> > Q: At the beginning of section 2.1 I do not understand the reference to word prediction and NLP.
> > A: Similar to the beginning of Section 2.2, the beginning of section 2.1 uses the most well-known example of RNN tasks (word prediction in NLP) and ConvNet task (feature extraction in computer vision) to define the notion of neighbourhood for these architectures. This is simply didactic - these examples are used as a first step to understand the extension of neighbourhood from regular grid (1D for NLP and 2D for computer vision) to arbitrary graphs (brain networks, social networks, etc) for RNNs and ConvNets.
> >
> > Q: ConvNets are more pruned to deep networks than RNNs
> > A: It simply means that graph ConvNets performance (with residuality) scales better than graph RNNs (with residuality).
> >
> > Q: What are "heterogeneous graph domains"?
> > A: Homogeneous graph domains refer to regular lattices and heterogeneous graph domains refer to graphs with complex variable structures like proteins, brain connectivity, gene regulatory network, etc.

---

### Official Review · AnonReviewer3 · 2017-11-27
**the relation between the two proposed models is not very clear**

**Rating:** 6
**Confidence:** 3

**Review:**

The paper proposes a new neural network model for learning graphs with arbitrary length, by extending previous models such as graph LSTM (Liang 2016), and graph ConvNets. There are several recent studies dealing with similar topics, using recurrent and/or convolutional architecture. The Related work part of this paper makes a good description of both topics.

I would expect the paper elaborate more (at least in a more explicit way) about the relationship between the two models (the proposed graph LSTM and the proposed Gated Graph ConvNets). The authors claim that the innovative of the graph Residual ConvNets architecture, but experiments and the model section do not clearly explain the merits of Gated Graph ConvNets over Graph LSTM. The presentation may raise some misunderstanding. A thorough analysis or explanation of the reasons why the ConvNet-like architecture is better than the RNN-like architecture would be interesting.

In the section of experiments, they compare 5 different methods on two graph mining tasks. These two proposed neural network models seem performing well empirically.

In my opinion, the two different graph neural network models are both suitable for learning graphs with arbitrary length,
and both models worth future stuies for speicific problems.

---

> ### Author Response · Authors · 2017-12-23
> **Author feedback**
>
> We thank the reviewer for her/his time and valuable comments. We hope to clarify any misunderstanding below and show the importance of this work in the field of deep learning on graphs.
>
> Q: Relationship between the two models
> A: There is no direct relationship between the proposed graph LSTM and the proposed graph ConvNet. We simply wanted to compare the best possible graph RNN architectures vs. the best graph ConvNets to find out what type of graph NNs should be used when dealing with problems involving graphs of variable length. As graph LSTM was not available in the literature, we simply used Scarselli-etal and Tai-etal to extend LSTM to arbitrary graphs. For the proposed graph ConvNet, we merged Sukhbaatar-etal and Marcheggiani-Titov, and added residuality to define the most possible generic ConvNet architecture for arbitrary graphs. Then, we performed several numerical experiments on graph matching and graph clustering to reach the conclusion that graph ConvNets should be preferred over RNN models for the class of variable graphs (such as molecules in quantum chemistry, gene regulatory networks for genetic disorders and particle physics for jet constituents).
>
> Q: Experiments on the merits of Graph ConvNets over Graph LSTM
> A: The most important advantage of graph ConvNets over graph LSTM is the multi-scale property. Graph ConvNet architectures have a monotonous increase of performance/accuracy when the network gets deeper, unlike RNN architectures for which performance decreases for a large number of layers. This property is illustrated in both graph experiments, see Figures 3 and 5 middle row. This makes ConvNet architectures more robust w.r.t. network design than RNN systems: Hyper-parameters such as L (nb of layers) and T (nb of inner RNN iterations, Fig4) must be carefully selected for graph RNNs, unlike graph ConvNets. Besides, RNN architectures are 1.5-4x slower than ConvNets (right column Figs 3 and 5) and they converge slower, Fig6.
>
> Q: Analysis why the ConvNet-like architecture is better
> A: We do agree such analysis would be important and we would like to carry it out in a future work. However, it is a challenging analysis as the data domain does not hold a nice mathematical structure like Euclidean lattices for images. This will require time and new analysis tools to develop such theory (given also that the standard theory for regular grids/images is still open).
> In the meantime, we hope the reviewer considers the rigorous numerical experiments - two fundamental graph experiments with controlled analytical settings (stochastic block models for the graph distributions) that offer a clear conclusion about graph ConvNets, which can be leveraged to build better NNs in the fast-emerging domain of deep learning on graphs.

---

### Decision · Program_Chairs · 2018-01-29
**ICLR 2018 Conference Acceptance Decision**

**Decision:**

Reject

**Comment:**

The authors make an experimental study of the relative merits of RNN-type approaches and graph-neural-network approaches to solving node-labeling problems on graphs.   They discuss various improvements in gnn constructions, such as residual connections.

This is a borderline paper.  On one hand, the reviewers feel that there is a place for this kind of empirical study, but on the other, there is agreement amongst the reviewers that the paper is not as well written as it could be.  Furthermore, some reviewers are worried about the degree of novelty (of adding residual connections to X).

I will recommend  rejection, but urge the authors to clarify the writing and expand on the empirical  study and resubmit.